# Energy Disaggregation Using Multi-Objective Genetic Algorithm Designed Neural Networks

**Inoussa Laouali [1,2]**, **Isaías Gomes [1,3,4]**, **Maria da Graça Ruano [1,5]**, **Saad Dosse Bennani [2]**, **Hakim El Fadili [6]** and **Antonio Ruano [1,3,*]**

1   DEEI, Faculty of Science & Technology, University of Algarve, 8005-294 Faro, Portugal
2   SIGER, Faculty of Sciences and Technology, Sidi Mohamed Ben Abdellah University, Fez P.O. Box 2202, Morocco
3   IDMEC, Instituto Superior Técnico, Universidade de Lisboa, 1950-044 Lisboa, Portugal
4   ICT, University of Evora, 7002-554 Evora, Portugal
5   CISUC, Faculty of Science & Technology, University of Coimbra, 3030-290 Coimbra, Portugal
6   LIPI, Faculty of Sciences and Technology, Sidi Mohamed Ben Abdellah University, Bensouda, Fez P.O. Box 5206, Morocco
*   Correspondence: aruano@ualg.pt

**Abstract:** Energy-saving schemes are nowadays a major worldwide concern. As the building sector is a major energy consumer, and hence greenhouse gas emitter, research in home energy management systems (HEMS) has increased substantially during the last years. One of the primary purposes of HEMS is monitoring electric consumption and disaggregating this consumption across different electric appliances. Non-intrusive load monitoring (NILM) enables this disaggregation without having to resort in the profusion of specific meters associated with each device. This paper proposes a low-complexity and low-cost NILM framework based on radial basis function neural networks designed by a multi-objective genetic algorithm (MOGA), with design data selected by an approximate convex hull algorithm. Results of the proposed framework on residential house data demonstrate the designed models' ability to disaggregate the house devices with excellent performance, which was consistently better than using other machine learning algorithms, obtaining F1 values between 68% and 100% and estimation accuracy values ranging from 75% to 99%. The proposed NILM approach enabled us to identify the operation of electric appliances accounting for 66% of the total consumption and to recognize that 60% of the total consumption could be schedulable, allowing additional flexibility for the HEMS operation. Despite reducing the data sampling from one second to one minute, to allow for low-cost meters and the employment of low complexity models and to enable its real-time implementation without having to resort to specific hardware, the proposed technique presented an excellent ability to disaggregate the usage of devices.

**Keywords:** non-intrusive load monitoring (NILM); energy disaggregation; neural networks; multi-objective genetic algorithm; low frequency power data; convex hull algorithms

## 1. Introduction

Since the start of the energy crisis, energy-saving problems have risen in popularity among the general public [1]. According to the study reported in [2], the global energy problem is driven by global population increase, rising demand, and continuous reliance on fossil fuels for production. Moreover, global energy consumption influences greenhouse gas emissions and, thus, climate change [3]. As a result of this situation, several governments have acted and set $CO^2$ emission reduction objectives. For instance, China, one of the world's major consumers of fossil fuels and major $CO^2$ emitter [2], promised to reach peak $CO_2$ emissions and carbon neutrality by 2030 and 2060, respectively [4]. Likewise, the European Union (EU) made a significant effort to combat global warming by implementing a variety of strategic policies [5,6].

The building sector is a prominent energy consumer and greenhouse gas emitter. For instance, it accounts for 27.5% of the overall final energy use in EU [5]. Therefore, there is a pressing need to halt the rising trend in building energy consumption. Energy management through adequate saving energy mechanisms has been the focus in recent years as a key strategy for conserving energy [1].

Home energy management systems (HEMS) are an efficient way to minimize energy consumption in a home while preserving occupant comfort [7,8]. The development of an effective HEMS system requires the implementation of a process to identify and monitor the principal loads in the household [9], thus allowing the HEMS to efficiently control the operation of (some) electric devices [10]. It also enables consumers to have a better awareness of their electricity usage patterns, potentially leading to more energy-conscious behavior and decreasing operating and electrical costs for grid operators and end users [11].

One of the techniques for monitoring appliances is to install measuring devices or sensors for each load of interest. This is known as intrusive load monitoring (ILM) in the literature and can properly estimate the operating condition of devices [12–14]. However, some disadvantages limit its practical application, such as the difficulties of deploying and configuring many sensors, its high cost, and its intrusive nature, which raises privacy and security concerns [15]. Non-intrusive load monitoring (NILM), on the other hand, aims to estimate the energy consumption of individual devices from a single meter that monitors the aggregated demand across many appliances [10]. It is one of the most effective energy disaggregation approaches since it enables users to disaggregate the power consumption of specific appliances in the household while maintaining user privacy and using smart meters that are already installed at the house entrance [15].

George Hart pioneered the use of NILM in [16,17]. He presented NILM as a monitoring system for appliances in an electrical circuit that turns ON and OFF independently. He showed that devices generate unique power consumption signatures. Therefore, the NILM techniques enable the recognition of these signatures from the total aggregated power consumption.

NILM algorithms are often classified into event-based algorithms [16–19] that relate signal state changes to device state changes and eventless algorithms that estimate a global system state using statistical and machine learning approaches [20]. Event-based approaches seek to detect and classify ON/OFF events of electrical devices. In Hart's approach, the ON/OFF events were used to detect the operation of specific appliances using the aggregated active and reactive powers. However, the technique struggled to identify certain types of appliances (multistate appliances, continuously variable consumer appliances, and permanent consumer devices).

Subsequently, several other techniques have been explored to solve the problem of energy disaggregation. Initially, researchers were drawn to the approaches of hidden Markov models (HMM) and their extensions [21–27]. However, as the number of target appliances increases, the complexity of HMM models and their extensions grow exponentially. Furthermore, difficulties with generalization and scalability were noted for these approaches, as well as a large sensitivity to current inference techniques for state estimation to local optima, thus limiting their applicability in the real world [13].

To tackle this NILM challenge, researchers have lately resorted to machine learning methods. Indeed, both supervised and unsupervised approaches have been explored to address the challenge of NILM. Support vector machines demonstrated good performance in [28–31]. K-nearest neighbors were explored in [32,33], decision tree in [34,35], k-means clustering in [36], and graph signal processing in [37,38]. More recently, deep learning approaches have prompted an upsurge in NILM research. The studies reported in [39,40] were among the first to use deep learning-based NILM. In [40], three deep learning models (long short-term memory, convolutional neural network, and autoencoders) were compared. The models were trained using a six-second low frequency sampling rate using only the active power as the input feature. They obtained better performance than the factorial HMM and combinatorial optimization state of the art models. In [41], the authors presented

a bidirectional long short-term memory model based on a low frequency sampling rate. They combined several electrical features to create a multi-feature input. The model was evaluated on low frequency data from the public datasets UKDALE and ECO. An approach based on deep convolutional networks was proposed in [42]. They classified the states of devices using a feature space enriched by the introduction of a temporal pooling module. The authors tested their model using the UKDALE low frequency dataset and demonstrated good generalization properties. In [43], a hybrid deep learning architecture based on a convex hull data selection approach using low frequency power data for NILM was proposed. They selected the most informative vertices of the real convex hull using a random approximation algorithm, incorporating them in the training data. They showed that the proposed framework is effective and performs well compared to existing approaches.

However, deep learning approaches require a large amount of training data to achieve satisfactory results [44]. This poses a major challenge for NILM algorithms due to the scarcity of high-quality datasets, both in terms of duration and label quality [10,45,46]. Additionally, such methods gain greatly from a large number of trainable parameters, which requires a processing capacity that is neither inexpensive nor cheaply accessible in most circumstances [44]. A comprehensive literature review of NILM approaches, beyond the scope of this paper, can be found in [15].

The application of NILM techniques depends strongly on the sampling rate, which refers to the frequency at which the meter measures the data. The sampling rate defines the type of information that can be obtained from the electrical signals [47]. There are two basic approaches for collecting data [13–15]: low frequency sampling rate (1 Hz or less) and high frequency sampling rate (in the range of kHz).

The high sampling frequencies enable the use of fine-grained characteristics such as harmonics [48], voltage-current (V-I) trajectory [49], and wavelet coefficients [50,51] from steady-state and transient. Although these methods may lead to good results in terms of device identification accuracy, high sampling frequency data have the drawbacks of being both difficult to transfer and store and very expensive in terms of software and hardware specificity [14]. Moreover, the current smart meter infrastructure, which typically allows for low sampling rates of the range of a few seconds, is not generally compatible with high frequency data collection and, therefore, specific equipment is required [52,53].

Conversely, low frequency methods are the ideal option in NILM applications since they enable the use of commonly used smart meter resources without requiring extra hardware. In principle, low-frequency data-based approaches perform load disaggregation by identifying the combination of states that substantially reduce the uncertainty margin [53]. However, the complexity and computational time of these methods may significantly increase with the number of appliances. Additionally, NILM algorithms must fulfill a variety of criteria, including good precision in recognizing device usage and accurately estimating their power consumption, using low complex models and low-cost equipment [15,41].

Therefore, there is the need to have an NILM technique available that, besides delivering very good results both in terms of operation detection and energy estimation, does not require specific acquisition systems, the existence of large amount of design data, and expensive processing power to design the classification and estimation models. To address these challenges, this paper proposes an NILM framework based on a low-frequency sampling rate, allowing the use of low-cost meters and employing low-complexity shallow neural network models. The following are the key contributions of this paper:

- A low-complexity NILM framework based on a radial basis function neural network designed by a multi-objective genetic algorithm (MOGA) with data selected by an approximate convex hull algorithm was proposed. The framework includes residential house data gathered in a real-life situation, feature extraction, appliance classification, and energy estimation.
- A comparative analysis of several computational intelligence's classifiers for non-intrusive load monitoring using the same data, including support vector machine,

k-nearest neighbors, decision trees, long short-term memory, and convolutional neural network, was performed.

- The feasibility of the approach was also validated using the public AMPds (Almanac of Minutely Power datasets) dataset, and a comparative study with other approaches was conducted.
- The proposed NILM framework was used to disaggregate one month of consumption of a case study house, identifying that 60% of the total consumption is related to schedulable devices, thus enabling a further degree of flexibility to HEMS.

The rest of the paper is structured as follows: Section 2 introduces the problem statement, the methods used, the data acquisition system, and the accuracy metrics employed. The obtained results are presented in Section 3, and their discussion is in Section 4. Section 5 concludes the paper.

## 2. Methodology

### 2.1. Problem Statement

The primary goal of NILM is to estimate the energy consumption of each individual device using aggregate data from the household's total energy usage. Let $x = [x_1, x_2, \ldots, x_T]$ be a sequence of aggregated readings from a smart meter in the house. The aim is to estimate the contribution of each device $y^{(n)} = [y_1^{(n)}, y_2^{(n)}, \ldots, y_T^{(n)}]$ in the house from $x$. This can be expressed in the following form:

$$x_t = \sum_{n=1}^{N} y_t^{(n)} + \varepsilon_t \tag{1}$$

where $\varepsilon_t$ represents the noise term, $N$ is the known number of appliances, $y_t^{(n)}$ is the contribution of appliance $n$ at time $t$, and $x_t$ is the aggregated data at time $t$.

The NILM process involves data collection, feature extraction, appliance classification, and energy estimation [14,15]. Figure 1 depicts the overview of the NLM framework. The initial step of an energy monitoring system is devoted to collecting aggregate data from the single meter. Following that, certain appliance features, or signatures, must be chosen and computed from the collected data. The classification method is carried out to identify which appliances are active at a given moment, as well as their states. Using this knowledge, the electric consumption of each device is estimated.

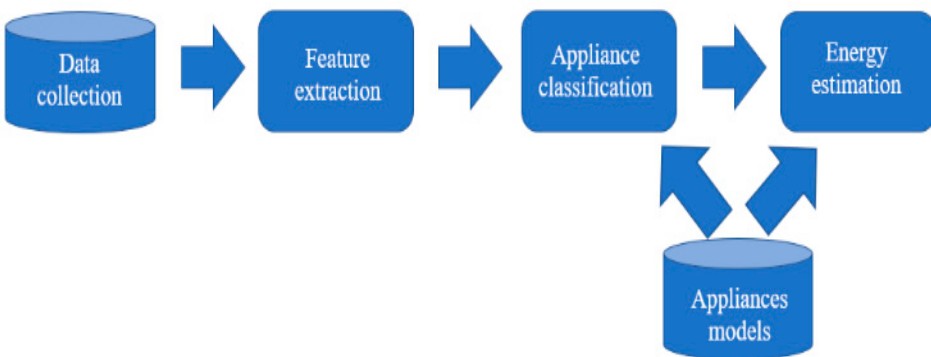

**Figure 1.** NILM overview.

In the literature, both supervised and unsupervised learning approaches are explored for appliance classification and energy estimation. However, in terms of accuracy, supervised techniques generally outperform unsupervised approaches [54]. In the proposed approach, appliance models were used in the accomplishment of these tasks.

The proposed NILM approach employed radial basis function neural networks designed by a multi-objective genetic algorithm (RBFNN-MOGA), with design data selected by an approximate convex hull algorithm. First, the Approxhull algorithm proposed in [55]

was applied to perform a data selection method based on the determination of the most informative vertices of the real convex hull. Afterwards, an RBFNN-MOGA classifier was designed to detect whether a specific device was ON or OFF. If the device was ON, then its power consumption was estimated using an RBFNN-MOGA estimator, designed in a similar manner to the RBFNN-MOGA classifier. Then, the models were designed, and the on-line operation is summarized in the Figure 2 flow chart.

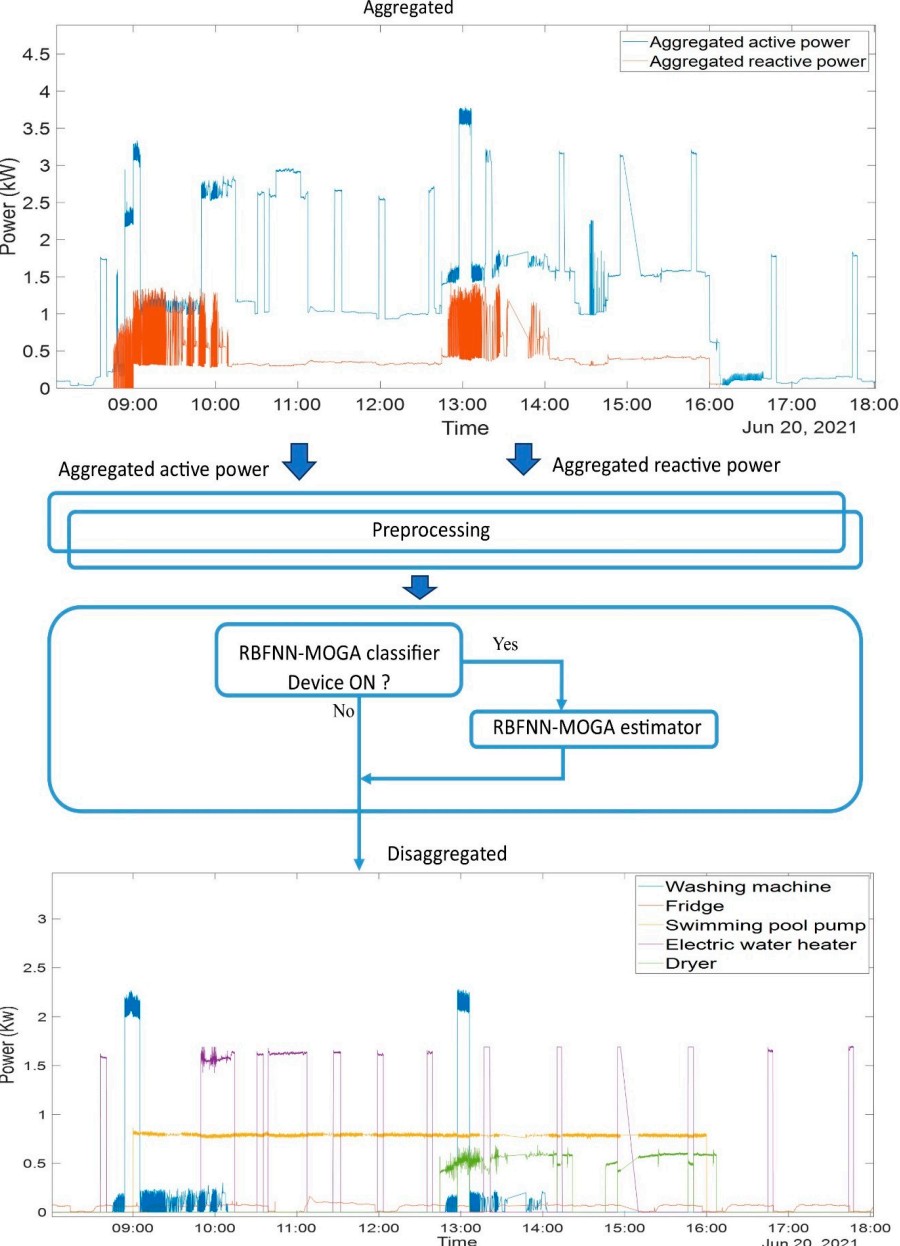

**Figure 2.** Overview of the proposed approach.

### 2.2. Data Collection

This study was based on data collected from a residential house located in Gambelas, Faro, Portugal. It has two floors and 20 separate spaces (including halls, a garden, rooms, and so on). A large variety of electrical devices were employed, and a json file was created using the NILM toolkit format [56]. The electric panel (Schneider) in the house includes 16 monophasic and 1 triphasic circuit breaker. Four smart plugs (SP), an intelligent weather station (IWS) [57], a PV installation with an inverter and a battery box [58], and four self-powered wireless sensors (SPWS) [59] are also available in the house. Using

this equipment, a data acquisition system was designed to keep track of several electric variables. A Carlo Gavazzi (EM340) [60] three-phase energy meter was used to collect the data needed for NILM algorithms. It monitored 45 different electric variables, which were all sampled at 1 Hz. Circutor Wibeees (CW) [61] were also employed to measure additional electric variables for every circuit breaker and to generate an approximate ground truth for NILM detection. Each one includes a hotspot for performing the first configuration via a manufacturer-supplied mobile app, and WB's data were sent to a free manufacturer web service by default. This procedure can be turned off, and the data collected can be accessed via an internal web interface, implementing the Modbus IP protocol. It is worth mentioning that the sampling instants for each breaker differ because the measurement equipment is not synchronized. The triphasic and the monophasic circuit breakers measure 198 variables every second, such as current, voltage, frequency, apparent, active, and reactive powers, the power factor, and capacitive reactive and active inductive energy. The four smart plugs allow for the measurement of six electric variables every second. The on/off control of certain equipment was performed using these smart plugs, and they also enabled us to measure individual sockets in the house. They can be accessed and controlled using a cloud API or directly through the internal web service. The intelligent weather station measures atmospheric variables such as air temperature, relative humidity, and global solar radiation. Data transmission from and to measuring appliances is performed using gateways and a technical network. The Modbus interface was used for data access using an IP-cabled and wireless network. An IoT platform was used both in the residence and in the cloud. It collected data from message queue servers that were configured. For each type of entity configured on the platform, the data were passed through a set of configured plug-ins. The system included a web page that allows the user to configure all the systems. Data can also be visualized using graphs organized by sensor type, and they can be downloaded in different formats (csv, xlsx, mat, and npz). The full description of the data acquisition system can be found in [62].

### 2.3. Data Preprocessing

The next step after data collection is the pre-processing of the datasets. Aggregated active and reactive powers were extracted from the EM340 meter data. The device's ground truth active power sequence was determined manually using CW and SP data. To create the ON-OFF labels of devices, an approach similar to the one proposed in [43] was used. Briefly, it was assumed that a device was switched ON when its power consumption exceeded a specified threshold value for at least a certain period of time. Table 1 summarizes the characteristics used to create the device labels, as well as basic statistics for each device, including the number of activations, the maximum and average activation durations, and the total active energy consumed, during the relevant data period. One dataset was created for each appliance.

**Table 1.** Devices statistics.

| Devices | Max Power (W) | Power Threshold (W) | Time Threshold (s) | Activations | ON Duration(s) | | Energy (kWh) |
|---|---|---|---|---|---|---|---|
| | | | | | Max | Average | |
| Fridge | 200 | 50 | 60 | 708 | 7045 | 2100 | 36.1 |
| Washing machine | 2500 | 20 | 3 | 17 | 8696 | 2250 | 9.5 |
| Electric water heater | 1700 | 1200 | 3 | 663 | 10,169 | 561 | 122.9 |
| Swimming pool pump | 1200 | 500 | 3600 | 28 | 25,202 | 24,985 | 159.1 |

### 2.4. Data Selection

For data selection, we used ApproxHull. In geometry, the convex hull of a set of data S can be considered the smallest convex set (region) including S. A set S is convex if, for every

couple $(u, v) \in$ S and any $t \in [0, 1]$, the point $(1 - t)u + tv$ is in S. Furthermore, if S is a convex set, for any $u_1, u_2, \cdots, u_j \in$ S, and any nonnegative numbers $\{\beta_1, \beta_2, \cdots, \beta_j\} : \sum_{i=1}^{j} \beta_i = 1$, the vector $\sum_{i=1}^{j} \beta_i u_i$ is called a convex combination of $u_1, u_2, \cdots, u_j$. The convex hull can be expressed as the intersection of all convex sets, including a given subset of Euclidean space, or similar to the set of all convex combinations of points in Euclidean space. It can be represented using vertices and facets, with vertices referring to the data set's border points and facets referring to the connections between the vertices.

ApproxHull is a data selection algorithm that uses a randomized convex hull approximation approach to process high-dimensional data in an optimal time and memory manner. It is an incremental algorithm that starts with an initial convex hull and then iteratively develops the current convex hull by introducing new vertices. In the approach, the vertices of the real convex hull were determined based on the hyperplane distance of the samples to the facets of the actual convex hull. It resulted in obtaining a subset of the most informative vertices of the real convex hull. For further description about the Approxhull algorithm, please refer to [55].

This way, using the design data, ApproxHull first removed duplicated samples and columns, retaining only the informative samples. The convex-hull points of this reduced set were mandatorily incorporated in the training set. According to user's specification, random points were extracted from the remaining design data to be added to the training set and to the testing and validation sets.

*2.5. Proposed Method*

2.5.1. Proposed Radial Basis Function Neural Networks Designed by Multi-Objective Genetic Algorithm (RBFNN-MOGA)

The radial basis function network is an artificial neural network made up of three fully connected layers: input layer, hidden layer, and output layer. The input layer consists of a set of inputs. It connects the source nodes to the hidden layer. The latter is made up of several units known as neurons. It performs a nonlinear transformation on the input data. Then, a linear combination of the hidden layer outputs is performed to generate the network's overall output. Equation (2) describes the output of the model [63,64]:

$$y(X) = \sum_{i=1}^{n} W_i \varphi_i(\|X - C_i\|) + b \tag{2}$$

where $W_i$ and $b$ denote the weights and bias term, $n$ is the number of neurons in the hidden layer, $\|.\|$ denotes the Euclidean norm, and $\varphi\{\|X - C_i\|\}_1^n$ are nonlinear radial basis activation functions centered at point $C_i$. The non-linear activation function used was the Gaussian function of the following Form (3):

$$\varphi(\|X - C_i\|) = e^{-\frac{1}{2\sigma_i^2}\|X - C_i\|^2} \tag{3}$$

where $\sigma_i$ denotes a spread parameter.

The design of the radial basis function neural network model was performed using a multi-objective optimization framework called MOGA. The method is a hybrid of a genetic algorithm and a derivative-based method. Its final result is a set of non-dominated models, obtained over a user-specified number of iterations. These individuals constitute the Pareto-optimal solutions and are obtained by minimizing predefined objectives. The genetic part searches the space of inputs and neurons, while the derivative algorithm estimates, for each individual model, its parameters.

Each potential neural network topology is expressed as a chromosome. The number of neurons in the hidden layer is the first element of the chromosome, and the following elements are the indices of an arbitrary number of features selected.

Let $D = (X, y)$ be a set consisting of $N$ input-output couples, which is split into three parts: training set $D_{tr}$, test set $D_{te}$, and validation set $D_{va}$. Let $L$ be a set of all possible input features. MOGA is supplied with dataset $D$, with the allowable range of hidden neurons $n \in [n_m, n_M]$ and the range of input features $d \in [d_m, d_M]$ from $L$ as design parameters. It then constructs a non-dominated collection of radial basis function neural network models that minimize $[\mu_p, \mu_s]$, where $\mu_p$ and $\mu_s$ represent a set of objectives associated with the radial basis function neural network's parameters $p$ and its structure, respectively. The model complexity, which is a function of the number of input features and the number of hidden neurons, is typically the only objective in $\mu_s$.

$$\mu_S = [O(\mu)] \tag{4}$$

Equation (5) describes the model complexity ($O(\mu)$):

$$O(\mu) = \mu_1 \times (\mu_2 + 1) \tag{5}$$

where $\mu_1$ and $\mu_2$ denote the number of neurons in the hidden layer and the number of input features, respectively.

In order to estimate the parameters of each model, a modified version of the Levenberg-Marquardt (LM) method was employed [65]. It exploits the neural network's linear/nonlinear separability property, typically obtaining a high accuracy and fast convergence rate.

As the models to be designed are nonlinear, the final results depend on the initial values of the parameters (centers and spreads). For this reason, the design framework allows different initialization methods, and instead of just training one model, a number of different models were trained for the same chromosome, and different strategies could be used to determine the best model for the current training trial.

Classification Problems

When used for classification problems, as is the case of the RBFNN-MOGA classifier, which detects if a device is ON or OFF, the relevant $\mu_p$ MOGA objectives are:

$$\mu_p = [FP_{D_{tr}}, FN_{D_{tr}}, FP_{D_{te}}, FN_{D_{te}}] \tag{6}$$

where $[FP_{D_{tr}}, FP_{D_{te}}]$ denote the false positives and $[FN_{D_{tr}}, FN_{D_{te}}]$ the false negatives for training and testing, respectively.

Estimation Problems

In the case of estimation problems, as is the case of the RBFNN-MOGA estimator that predicts the device power consumption, a nonlinear autoregressive with exogenous inputs (NARX) configuration was employed. The specification of $\mu_p$ was based on minimizing the error between model outputs and target values. The $\mu_p$ objectives were as follows:

$$\mu_p = [\varepsilon(D_{tr}), \varepsilon(D_{te}), \varepsilon(D_s, PH)] \tag{7}$$

where $\varepsilon(D_{tr})$ and $\varepsilon(D_{te})$ represent the root mean square errors (RMSE) of the model, assuming the training set $D_{tr}$ and the testing set $D_{te}$. $\varepsilon(D_s, PH)$ denote the forecasting error. It is computed by summing the RMSEs along the prediction horizon ($PH$):

$$\varepsilon(D_s, PH) = \sum_{i=1}^{PH} RMSE(E(D_s, PH), i) \tag{8}$$

$E$ is an error matrix defined as:

$$E(D_s, PH) = \begin{bmatrix} e[1,1] & e[1,2] & \cdots & e[1,PH] \\ e[2,1] & e[2,2] & \ddots & e[2,PH] \\ \vdots & \vdots & \cdots & \vdots \\ e[p-PH,1] & e[p-PH,2] & \vdots & e[p-PH,PH] \end{bmatrix} \tag{9}$$

where $e[i,j]$ denotes the model prediction error taken from instant $i$ of $D_s$ at step $j$ within the prediction horizon $PH$. $D_s$ is a time series with $p$ data points.

Figure 3 depicts a flowchart illustrating the processes involved in the model's design. This approach was used in the literature to solve several classification and prediction problems [64,66,67]. A full description of the MOGA design radial basis function neural network can be found in [68].

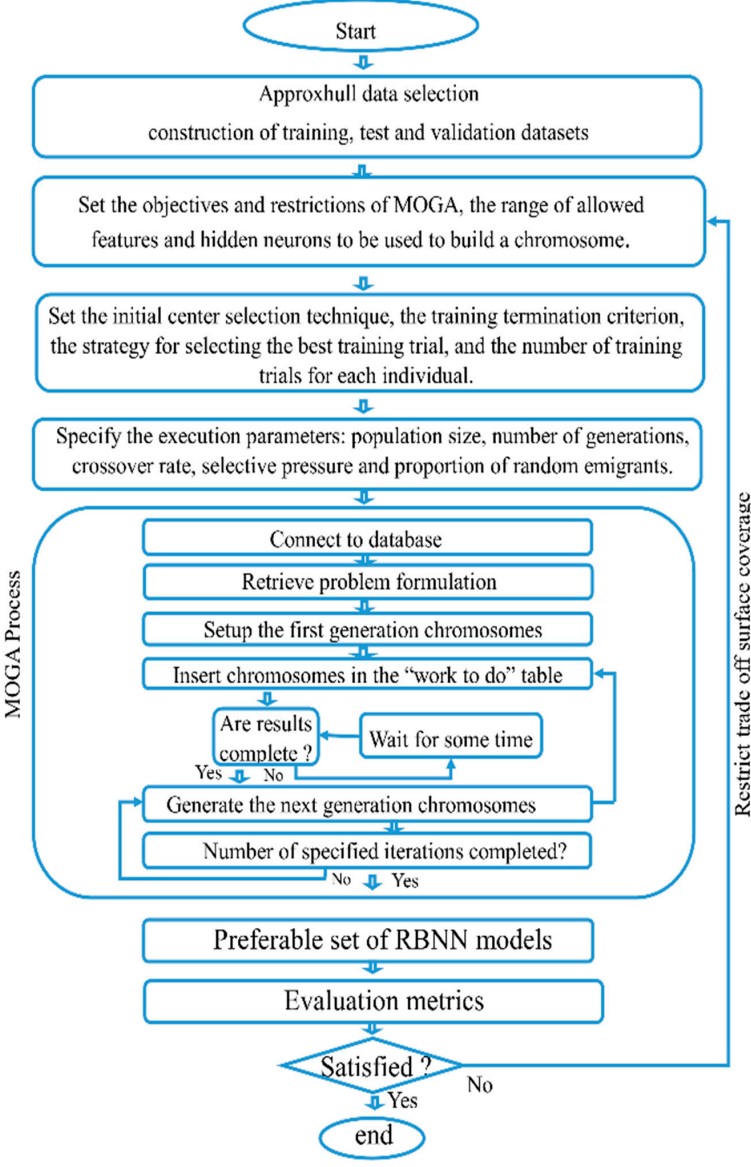

**Figure 3.** Overview of the design approach.

### 2.5.2. Other Implemented Classification Methods

To assess the performance of the proposed method, different classification alternatives were implemented:

Support Vector Machines (SVM)

Support vector machines are a class of supervised learning algorithms used to solve classification and regression problems. They emerged from Vladimir Vapnik's theoretical considerations on the development of a statistical theory of learning in the 1990s [69]. In the case of a classification problem, the goal of the SVM is to find the maximum margin hyperplane to separate the positive examples from the negative ones. Given a training set $[x_i, y_i]_{i=1}^N$, where $x_i \in \mathbb{R}^d$ is the $i$th vector of the input features and $y_i \in \mathbb{R}$ is the target, the support vector method aims to create a classifier described as follows [70]:

$$o(x_i) = \text{sign}\left[\sum_{i=1}^{n} \alpha_i \Psi(x, \ x_i) + b\right] \tag{10}$$

$$\Psi(x, \ x_i) = e^{-\frac{1}{2\sigma_i^2}\|x-x_i\|^2} \tag{11}$$

$$y_i\left[w^T \zeta(x_i) + b\right] \geq 1 \tag{12}$$

$$\zeta^T(x)\zeta(x_i) = \Psi(x, x_i) \tag{13}$$

where $n$ is the number of support vectors, $\zeta(.)$ is a nonlinear function that maps the input space into a high-dimensional space. $\Psi$ is the RBF kernel function used, $b$ is a real constant, $\alpha_i$ are positives real constants, and $\sigma$ is a spread parameter.

K Nearest Neighbors (KNN)

The nearest neighbors' algorithm is a non-parametric supervised classification algorithm [71]. The approach estimates the output associated with a new input $x$ by considering the k samples whose input is closest, in a Euclidean sense, to the new input $x$. The choice of k can lead to classification issues because the results are sensitive to noise if k is too small, and the precision is lowered if k is too large. In the classification phase, k is a user-defined constant, and an unlabeled vector is classified by assigning it the label that appears the most often among the k training samples. The distance d is determined by using (14) for any instance $x_i$ in a data set of size N [72].

$$d(x_i) = \min\{d(x_i, x_1), \ d\ (x_i, x_2), \ \ldots, \ d(x_i, x_n)\} \tag{14}$$

Euclidean distance was used in this work. Equation (15) calculates this distance.

$$d_E(\{x_1, x_2, \ \cdots, x_N\}; \{y_1, y_{2,\cdots}, y_N\}) = \sqrt{\sum_1^N (x_i - y_i)^2} \tag{15}$$

where $N$ is the total number of dimensions.

Decision Trees

A decision tree's structure can be viewed as a tree-like representation of the decision process [35]. It classifies the data based on the tree structure's properties, with a leaf node representing a record set subject to a constraint. Quilan [73] introduced the ID3 algorithm for decision tree classification in 1986. Subsequently, the C4.5 method was created as a result of the ID3 algorithm's inability to deal with discrete attributes and its tendency to over-fit. It was based on the automatic selection of discriminant variables from unstructured and potentially large data. Decision trees can thereby infer logical

cause-and-effect principles that did not initially appear in the raw data. The numerous possible decisions are established at the branches' ends and are closely aligned with the decisions taken at each level. As was described in [35], the definition of the C4.5 algorithm is as follows: A training set is made up of A data points, with the form $(V_1, V_2, \cdots, V_a, C)$, where $V_i$ is the sample's attribute value and C is the sample's category. Assuming the set of samples $T$ is partitioned into subsets $T_1, T_2, \cdots, T_k$ based on the discrete attribute $A$'s k distinct values, the information *Gain* is used to determine the splits and is calculated as follows:

$$Gain\ (A,\ T) = \inf(T) - \sum_{i=1}^{k} \frac{|T_i|}{|T|} \times \inf\ (T_i) \tag{16}$$

Long Short-Term Memory (LSTM)

LSTM is a deep recurrent neural network that processes sequential data and deals with vanishing gradient problems using a feed-forward variation. It was developed by Hochreiter [74] in order to learn long short-term dependency information. Its architecture is based on recursive state calculation using prior states and inputs. An LSTM layer is made up of memory blocks, which are recurrently connected blocks. There are one or more recurrently connected memory cells in each block. For the cells, the forget layer, input layer, and output layer are three multiplicative units that enable the continuous analogues of write, read, and reset processes [75]. The forget layer decides which information should be saved and which should be eliminated. Its output is computed utilizing the current input sample's weight and bias parameters, as well as data from the previous time step. The forget gate, input gate, update gate, and output gate are all part of the gate mechanism. The LSTM model's mathematical formula is as follows [76]:

$$z^t = \varphi(W_Z x^t + R_z y^{t-1} + b_z) \tag{17}$$

$$i^t = \sigma\left(W_i x^t + R_i y^{t-1} + p_i \odot c^{t-1} + b_i\right) \tag{18}$$

$$f^t = \sigma\left(W_f x^t + R_f y^{t-1} + p_i \odot c^{t-1} + b_t\right) \tag{19}$$

$$c^t = z^t \odot i^t + c^{t-1} \odot f^t \tag{20}$$

$$o^t = \sigma\left(W_O x^t + R_O y^{t-1} + p_o \odot c_t + b_o\right) \tag{21}$$

$$y^t = \zeta\left(c^t\right) \odot o^t \tag{22}$$

where $\sigma$, $\varphi$, and $\zeta$ are point-wise nonlinear activation functions, $\odot$ denotes the vector pointwise multiplications, and input, recurrent, peephole, and bias weights are denoted by $W$, $R$, $p$, and $b$, respectively. Further information about LSTM models can be found in [74].

Convolutional Neural Network (CNN)

CNN is one of the most widely used deep learning networks in large-scale image recognition due to its success in classification tasks [77,78]. CNN architecture stems from the combination of several convolutional layers, pooling layers, and fully connected layers. The convolutional layer enables the extraction of features from the input; the extracted features are then selected and filtered by the pooling layer, and the output is generated by nonlinearly combining the selected features with the fully connected layer [79]. Given a time-dependent signal $x(t)$ and a kernel $(k)$ of size $K$, the cross-correlation process generates another signal $y(t)$, which may be computed as follows [80]:

$$y(t) = \sum_{k=-\left[\frac{K}{2}\right]}^{\left[\frac{K}{2}\right]} x(t+k)\, k(k) + b \tag{23}$$

where $b$ denotes the bias term. Following that, the result is passed to the *ReLu* activation function used.

$$ReLu(x) = max\,(0, x) \tag{24}$$

The output of the fully connected layer is defined by:

$$y = ReLu\,(wx + b) \tag{25}$$

where $w$ denotes the layer's weight matrix, and $b$ denotes the layer's bias vector.

### 2.6. Performance Metrics

Several metrics are used in the literature to evaluate NILM algorithms [42,45,81]. To assess the performance of the proposed models, the most relevant ones were selected with the goal of capturing the methodology's efficiency in both identifying the activation state and estimating the energy usage.

For classification, the *recall* metric, which assesses the model by scoring particular predictions about the expected data as positive, the *precision*, which is defined as the proportion of relevant instances found among the retrieved instances, and *F1*, i.e., the harmonic mean of *recall* and *precision*, are the criteria employed. The first two criteria depend on the number of true positives (*TP*), which denotes the number of properly predicted testing periods when the target appliance was ON, on the number of false positives (*FP*), which is the number of OFF periods for the target appliance that are erroneously identified as ON, and on false negatives (*FN*), which is the number of ON periods that were mistakenly categorized as OFF.

Regarding energy estimation, the mean absolute error (MAE) assesses the accuracy of the estimate power consumption of device, the signal aggregate energy (SAE) measures the relative error in estimating the amount of energy consumed during the full assessment period, and the estimation accuracy (EA) reflects how well the NILM algorithm estimates power usage in comparison to real consumption.

$$Recall = \frac{TP}{TP + FN} \tag{26}$$

$$Precision = \frac{TP}{TP + FP} \tag{27}$$

$$F1 = 2 \times \frac{Precision \times Recall}{Precision + Recall} \tag{28}$$

$$MAE^{(k)} = \frac{\sum_1^N \left| y_t^k - p_t^k \right|}{N} \tag{29}$$

$$SAE^{(k)} = \frac{\left| \hat{E}^{(k)} - E^{(k)} \right|}{E^{(k)}} \tag{30}$$

$$EA^{(k)} = 1 - \frac{\sum_1^N \left| y_t^k - p_t^k \right|}{2 * \sum_1^N y_t^k} \tag{31}$$

where $N$ is the number of values that represent the load data, $y_t^k$ are the ground truth values, $p_t^k$ are the predicted values, $\hat{E}^{(k)}$ is the total predicted energy, and $E^{(k)}$ is the total ground truth energy for device $k$. All these metrics are dimensionless, except MAE, for which the unit of measurement is Wh.

## 3. Results

This section presents the results of tests performed on the data collected in the house presented in Section 2.4. The data were collected for nearly two years and can be found in (https://csi.ualg.pt/nilmforihem, accessed on 1 November 2022). The experiments

were conducted using data collected over a one-month period, from 1 March 2021 to 28 March 2021 for the washing machine and from 2 June 2021 to 28 June 2021 for the other devices, totaling approximately 2.4 million samples for each device. Since the original data were collected with a sampling interval of one second, their size was too large to be processed by the MOGA framework and was therefore resampled at a 1 min rate. The proposed framework was evaluated using four popular devices for evaluating NILM algorithms, namely a fridge, washing machine, electric water heater, and swimming pool pump. Moreover, these devices are within the range of the main energy consuming devices in the studied house.

### 3.1. MOGA Design Radial Basis Function Neural Network Results

The next step is to perform data selection before applying MOGA to design the neural networks. Indeed, in order to design the suitable final solution, the system must train a significant number of radial basis function neural network models. On the one hand, certain constraints should be implemented on the size of the datasets that are supplied to MOGA to enable the procedure to be completed in an acceptable time period and, on the other hand, to ensure the quality of the data to be used for the training set. To achieve this, the Approxhull algorithm described in Section 2.4 was applied. First, the original dataset containing the target and the features was provided to the algorithm. For the state classification of the devices, the input features were a sliding window consisting of 20 variables (10 aggregated active and 10 reactive delayed power values). For the power estimation, lags of two exogenous variables (aggregated active and reactive powers, 10 for each variable) as well as 10 lags of the modeled variable (appliance active power) were fed to the ApproxHull algorithm.

Using ApproxHull, the convex hull points of all relevant data samples were generated. The MOGA training set was then built using the obtained convex points and random data samples to obtain 60% of the data. MOGA testing sets and validation sets were built using the reminder random data samples with a proportion of 20% for each part. Moreover, it should be noted that before computing the convex-hull (CH) vertices, the original data set went through a cleaning phase. Tables 2 and 3 present the number of CH vertices found and the size of training, testing, and validation sets generated for each appliance, for classification and for estimation, respectively.

**Table 2.** Approxhull results for classification.

| Devices | CH Vertices | Training | Testing | Validation |
| --- | --- | --- | --- | --- |
| Fridge | 441 | 24,493 | 8164 | 8166 |
| Washing machine | 810 | 24,185 | 8061 | 8063 |
| Electric water heater | 731 | 8212 | 2737 | 2739 |
| Swimming pool pump | 673 | 12,655 | 4218 | 4220 |

**Table 3.** Approxhull results for estimation.

| Devices | CH Vertices | Training | Testing | Validation |
| --- | --- | --- | --- | --- |
| Fridge | 842 | 24,185 | 8061 | 8063 |
| Washing machine | 753 | 24,185 | 8061 | 8063 |
| Electric water heater | 1497 | 8212 | 2737 | 2739 |
| Swimming pool pump | 640 | 12,655 | 4218 | 4220 |

For classification and for each appliance, the MOGA algorithm was executed with the objective of minimizing the model complexity, the number of false positives and negatives in the training set, and the number of false positives and negatives in the testing set.

For estimation, MOGA minimized the RMSEs of the training set $\varepsilon(D_{tr})$ and testing set $\varepsilon(D_{te})$, the model complexity, and the forecasting error $\varepsilon(D_s, PH)$. The prediction horizon

(*PH*) was set to 1, and $D_s$ represents a specific period extracted from the time-series of the appliance active power (with s consecutive input-output pairs).

According to the author's experience with using MOGA, the algorithm parametrization was as follows. For both models, the hidden layer's number of neurons was set in the range of [2,30], whereas the number of input features was set in the range of [1,20] for the classification and in the range of [1,30] for the estimation models. A population size of 100 was used, with a number of generations fixed at 100. The proportion of random immigrants was set to 10%. The crossover rate was set to 0.7, with a selective pressure of 2. The maximum number of iterations was set to 50. Each individual in the population was trained 10 times with different initial conditions, and the best training trial was selected using the nearest to the origin criterion. An early stopping criterion was used as the termination criterion.

After one run of MOGA, non-dominated sets of models were generated. Table 4 presents the dimensions of the non-dominated sets for classification and estimation for each appliance.

**Table 4.** Dimensions of non-dominated sets.

| Devices | Classification | Estimation |
|---|---|---|
| Fridge | 274 | 45 |
| Washing machine | 89 | 171 |
| Electric water heater | 414 | 329 |
| Swimming pool pump | 153 | 172 |

Once the models were designed, the classification model was primarily employed to detect if the appliance was active or not in the next sample. If the device was active, then the MOGA-designed estimator model was employed to estimate the power consumption of the device. Table 5 presents the performance statistics in terms of minimum and average false positives (*FP*) and false negatives (*FN*) in the three sets, as well as the model complexity obtained in the classification phase for each appliance. As can be seen, the number of errors was small compared to the dimensions of the sets.

**Table 5.** Classification performance in the non-dominated sets.

| Devices | | Training | | Testing | | Validation | | Model Complexity |
|---|---|---|---|---|---|---|---|---|
| | | *FP* | *FN* | *FP* | *FN* | *FP* | *FN* | |
| Fridge | Min | 1722 | 0 | 534 | 0 | 550 | 0 | 6 |
| | Mean | 2638 | 295 | 843 | 96 | 836 | 103 | 139 |
| Washing machine | Min | 0 | 20 | 0 | 10 | 0 | 7 | 8 |
| | Mean | 24 | 220 | 4 | 56 | 6 | 55 | 282 |
| Electric water heater | Min | 58 | 1 | 24 | 0 | 25 | 0 | 6 |
| | Mean | 231 | 231 | 79 | 68 | 60 | 71 | 162 |
| Swimming pool pump | Min | 6 | 0 | 2 | 0 | 2 | 0 | 6 |
| | Mean | 84 | 49 | 23 | 14 | 23 | 12 | 143 |

Performance statistics in terms of the minimum and mean values of the root mean square errors for the training, testing, and validation sets, evaluated in the non-dominated sets in the estimation phase, are presented in Table 6.

**Table 6.** Estimation performance in the non-dominated sets.

| Devices | | Training | Testing | Validation |
|---|---|---|---|---|
| Fridge | $\epsilon_{min}$ | 0.131 | 0.124 | 0.133 |
| | $\bar{\epsilon}$ | 0.136 | 0.129 | 0.136 |
| Washing machine | $\epsilon_{min}$ | 0.010 | 0.018 | 0.010 |
| | $\bar{\epsilon}$ | 0.031 | 0.032 | 0.025 |
| Electric water heater | $\epsilon_{min}$ | 0.250 | 0.221 | 0.239 |
| | $\bar{\epsilon}$ | 0.298 | 0.260 | 1.710 |
| Swimming pool pump | $\epsilon_{min}$ | 0.063 | 0.048 | 0.057 |
| | $\bar{\epsilon}$ | 0.069 | 0.048 | 1.790 |

After some further tests, for each appliance and each model type, the non-dominated sets' best model, with high performance and low complexity, was selected. Table 7 presents the classification results of the validation dataset using the selected model in terms of *recall*, *precision*, *F*1 score, number of features, number of neurons in the hidden layer, and model complexity for each device. As can be seen, models with very low complexity obtained excellent classification results.

**Table 7.** Classification results.

| Devices | R | P | F1 | Number of Features | Number of Neurons | Model Complexity |
|---|---|---|---|---|---|---|
| Fridge | 0.98 | 0.91 | 0.95 | 7 | 30 | 240 |
| Washing machine | 0.97 | 0.97 | 0.97 | 4 | 30 | 150 |
| Electric water heater | 0.99 | 0.97 | 0.98 | 12 | 29 | 377 |
| Swimming pool pump | 1 | 1 | 1 | 8 | 25 | 225 |

R: Recall, P: Precision.

The estimation results using the selected models in terms of mean absolute error (MAE), signal aggregate error (SAE), and estimation accuracy (EA) are presented in Table 8.

**Table 8.** Estimation results.

| Devices | MAE (W) | SAE | EA | Number of Features | Number of Neurons | Model Complexity |
|---|---|---|---|---|---|---|
| Fridge | 4.00 | 0.010 | 0.96 | 4 | 6 | 30 |
| Washing machine | 1.1 | 0.030 | 0.93 | 23 | 20 | 480 |
| Electric water heater | 66.7 | 0.043 | 0.95 | 27 | 18 | 504 |
| Swimming pool pump | 1.5 | 0.004 | 0.99 | 7 | 14 | 112 |

The results presented in Tables 7 and 8 illustrate the model's efficiency in detecting and estimating the energy consumed by each appliance. The detection results of *F*1 scores were over 95% for fridge, 97% for washing machine, 98% for electric water heater, and 99% for swimming pool pump. In terms of estimation accuracy, the washing machine obtained over 93%, the electric water heater 95%, the fridge 96%, and the swimming pool pump 99%. The EA results were all above 0.93, again using models of very small complexity.

Figures 4–7 depict examples of disaggregation outputs in terms of active power. For all figures, the blue line denotes the ground truth values, and the red line denotes the estimated ones.

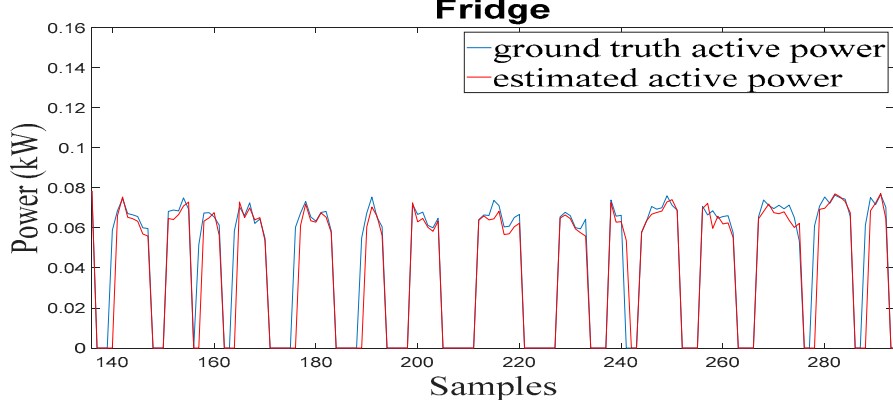

**Figure 4.** Fridge: blue, ground truth active power; red, estimated active power.

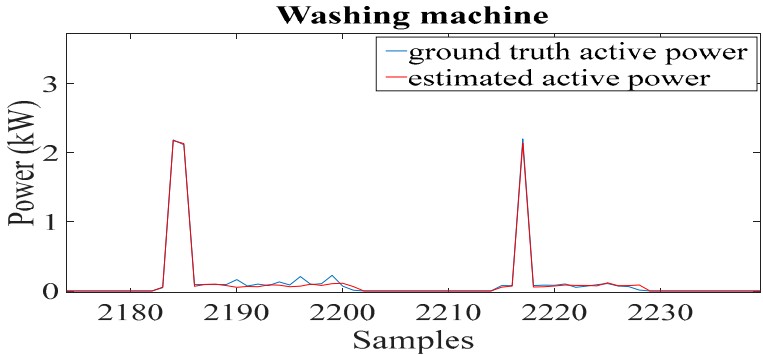

**Figure 5.** Washing machine: blue, ground truth active power; red, estimated active power.

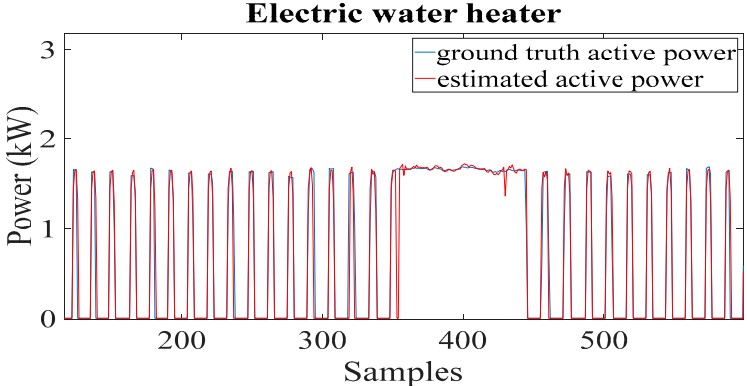

**Figure 6.** Electric water heater: blue, ground truth active power; red, estimated active power.

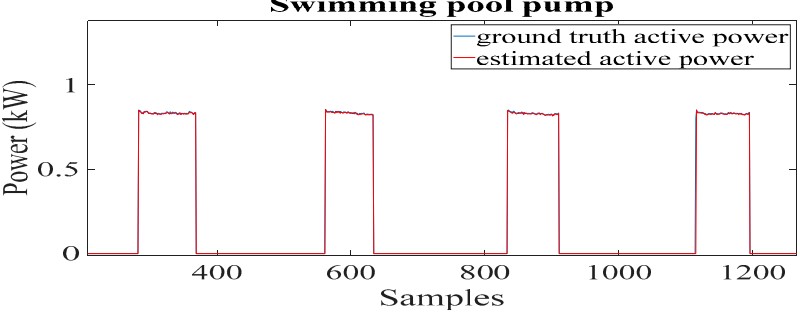

**Figure 7.** Swimming pool pump: blue, ground truth active power; red, estimated active power.

As can be observed, there was a consistent agreement between the measured and the predicted active power for the devices considered in the study. The predicted power consumption signals were highly reliable. From these results, we concluded that the proposed method achieved a very satisfactory performance, with models of very low complexity operating on one-minute data.

### 3.2. Results of Other Implemented Classification Methods

Utilizing the same 1 month of data, five experiments were conducted using different classification methods such as support vector machines, k-nearest neighbors, decision trees, LSTM, and CNN. The first step was to standardize the data to accelerate the training process and to make the training procedure more robust. The standardization of aggregated data was conducted in a way similar to the one reported in [40]. First, each sequence's mean and standard deviation were determined. The normalized sequence was then created by subtracting the mean from the input sequence and dividing it by the standard deviation. The goal was to have a normalized sequence with a standard deviation of one and a mean of zero. Equation (32) describes the procedure:

$$X_{\text{normalized}} = \frac{(X - \mu)}{\sigma} \sim N(0,1) \tag{32}$$

where $X$ is the input sequence, $\mu$ is the mean of the input sequence, and $\sigma$ is the standard deviation of the input sequence.

The dataset was split into a training set (80%) and testing set (20%). As in the proposed framework, a sliding window of 20 variables (10 aggregated active and 10 aggregated reactive power values) was used.

#### 3.2.1. Support Vector Machines Results

The model was trained to classify the state of the device. The purpose was to determine whether or not the appliance was active. One model was trained for each device. The gaussian radial basis function kernel was used. The penalty C and spread parameters σ were adjusted to train the model by minimizing the error function [69] defined in (33):

$$error\ (x_i, y_i) = \frac{1}{2} w'w + C \sum_{i=1}^{N} \xi_i \tag{33}$$

where $C$ denotes the penalty parameter, $w$ represents the weights vector, and $\xi_i$ denotes the slack variables. Automatic hyperparameter optimization was used to find hyperparameters that minimized the five-fold cross-validation loss. The results obtained in the test dataset in terms of recall, precision, and *F*1-score are presented in Table 9.

**Table 9.** SVM results.

| Devices | *Recall* | *Precision* | *F*1 |
|---------|----------|-------------|------|
| Fridge | 0.91 | 0.90 | 0.91 |
| Washing machine | 0.91 | 0.75 | 0.82 |
| Electric water heater | 0.98 | 0.96 | 0.97 |
| Swimming pool pump | 0.98 | 0.99 | 0.98 |

#### 3.2.2. K Nearest Neighbors (KNN) Results

As for SVM, the model was trained to classify the state of the device. One model was trained for each appliance. Since k is a user-defined parameter, a trial-and-error procedure was conducted to choose the optimal k that fit the data. The best value found was k = 5. The results obtained in the test dataset in terms of recall, precision, and *F*1-score are presented in Table 10.

**Table 10.** KNN results.

| Devices | *Recall* | *Precision* | *F1* |
|---|---|---|---|
| Fridge | 0.90 | 0.92 | 0.91 |
| Washing machine | 0.91 | 0.73 | 0.81 |
| Electric water heater | 0.98 | 0.95 | 0.97 |
| Swimming pool pump | 0.99 | 0.99 | 0.99 |

### 3.2.3. Decision Tree Results

Such as SVM and KNN, the decision tree classifier was trained to classify the state of the devices. One model was trained for each device. A cross-validation method was applied, consisting of randomly dividing the training data into ten subsets. On each of the nine subsets of the data, ten new trees were trained. The predictive performance of each new tree was then evaluated using data that were not used for its training. A good estimate of the predictive accuracy of the resulting tree was thus obtained. The automatic optimization of hyperparameters was used to find the optimal value for the minimum number of observations of leaf nodes (MinLeafSize) that minimized the cross-validation loss. Table 11 presents the results obtained in the test dataset in terms of recall, precision, and *F*1-score.

**Table 11.** Decision tree results.

| Devices | *Recall* | *Precision* | *F1* |
|---|---|---|---|
| Fridge | 0.87 | 0.93 | 0.90 |
| Washing machine | 0.94 | 0.73 | 0.82 |
| Electric water heater | 0.98 | 0.93 | 0.95 |
| Swimming pool pump | 0.98 | 0.99 | 0.98 |

### 3.2.4. Long Short-Term Memory (LSTM) Results

An LSTM classifier was built to classify the state of the devices. An Adam optimizer and binary cross entropy loss function were used for training the model. The best fitting set of parameters was found by minimizing the loss function through a trial-and-error procedure. Hyperparameter optimization was used to find the hyperparameters that minimized the five-fold cross-validation loss. Table 12 summarizes the architecture used with the optimal hyperparameter values. One model was trained for each device.

**Table 12.** LSTM structure.

| Layer | Number of Hidden Units | Activation |
|---|---|---|
| Input | | |
| LSTM | 32 | ReLu |
| Dropout (dropout = 0.3) | | |
| LSTM | 64 | ReLu |
| Dropout (dropout = 0.3) | | |
| LSTM | 128 | ReLu |
| Dropout (dropout = 0.3) | | |
| Fully connected dense | 1024 | ReLu |
| Fully connected dense | 1 | sigmoid |

The results obtained in the test dataset in terms of recall, precision, and F1 score are presented in Table 13.

**Table 13.** LSTM results.

| Devices | Recall | Precision | F1 |
|---|---|---|---|
| Fridge | 0.89 | 0.83 | 0.86 |
| Washing machine | 0.82 | 0.96 | 0.88 |
| Electric water heater | 0.98 | 0.94 | 0.96 |
| Swimming pool pump | 0.97 | 0.99 | 0.98 |

### 3.2.5. Convolutional Neural Network (CNN) Results

As for LSTM, a CNN classifier was built. The model was trained using an Adam optimizer and binary cross entropy loss function. A trial-and-error procedure was used to find the best-fitting set of parameters by minimizing the loss function. The hyperparameters that minimized the five-fold cross-validation loss were found using hyperparameter optimization. The architecture employed with the optimal hyperparameter settings is summarized in Table 14. One model was trained for each device.

**Table 14.** CNN structure.

| Layer | Filters | Kernels | Activation |
|---|---|---|---|
| Input | | | |
| Conv1D | 32 | 3 | ReLu |
| Conv1D | 64 | 3 | ReLu |
| Conv1D | 128 | 3 | ReLu |
| Max pooling1D | | | |
| dense (1024) | | | ReLu |
| dense (1) | | | sigmoid |

Table 15 presents the results of the tests performed on the test dataset in terms of *recall*, *precision*, and *F*1 score.

**Table 15.** CNN results.

| Devices | Recall | Precision | F1 |
|---|---|---|---|
| Fridge | 0.92 | 0.91 | 0.91 |
| Washing machine | 0.96 | 0.90 | 0.93 |
| Electric water heater | 0.95 | 0.98 | 0.96 |
| Swimming pool pump | 0.99 | 0.99 | 0.99 |

## 4. Discussion

By analyzing the results presented in Tables 7, 9–11, 13 and 15, we observed that all the models explored can detect the operating states of the different devices with good performance. To illustrate the effectiveness of the proposed method, the *F*1-score value achieved with the proposed method was compared with the other state-of-the-art classification methods. Figure 8 presents a comparative histogram of the different methods implemented.

For fridge, the best *F*1 score of 95% was achieved by the proposed technique, while LSTM achieved the worst *F*1 score of 86%. The other models classified the fridge with a similar *F*1 score of 91% for SVM, KNN, and CNN and 90% for DT.

For washing machine, the proposed model obtained the best *F*1 score of 97%, while the SVM and decision tree models obtained similar *F*1 scores of 82%. Similarly, the two deep learning models obtained slightly better results than the SVM and decision tree models, with the CNN obtaining an *F*1 score of 93% and the LSTM an *F*1 score of 88%. The KNN model achieved the worst *F*1 score of 81%.

For the electric water heater, the proposed approach obtained an *F*1 score of 98% slightly higher than the *F*1 score of 97% obtained by the SVM and KNN models. The LSTM

and CNN models had a comparable *F*1 score of 96%, and the decision tree model had the lowest *F*1 score of 95%.

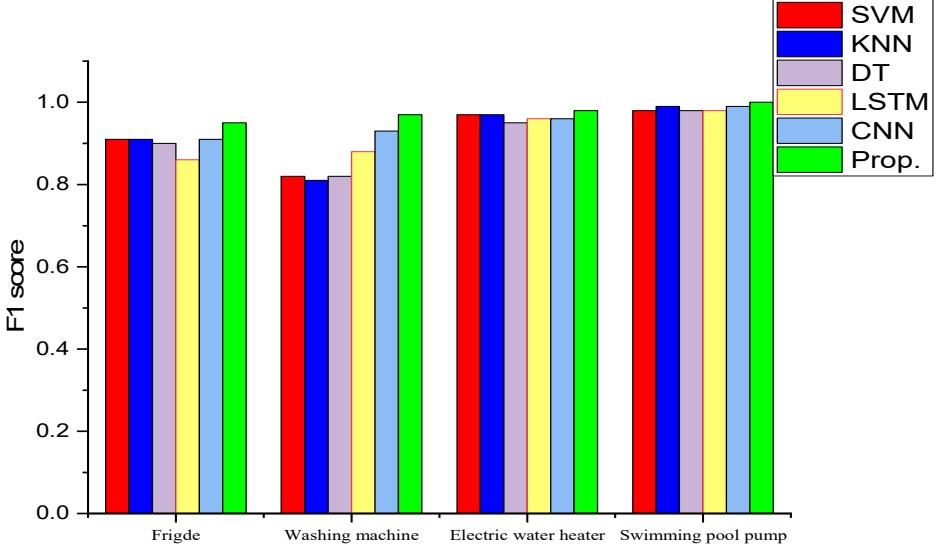

**Figure 8.** *F*1 comparison.

All the models performed well in the classification of the swimming pool pump, with an *F*1 score of more than 98% for the models SVM, KNN, decision tree, CNN, and LSTM and 100% for the proposed model. As can be seen, the best performances were obtained for high consumption devices, while most of the algorithms seemed to have more difficulties correctly detecting the washing machine and fridge activations. This is due to the complex architecture of these multi-state appliances, for which detection is a little more challenging for some models, resulting in a large number of false positives. Overall, it can be seen that the proposed framework achieved the best performance in terms of *F*1 score.

The performance of the six methods presented earlier was qualitatively comparable, as the same data were used for each one of the techniques. This is typically not the case when comparing the performance of state-of-the-art approaches found in the literature, as the data used and the context of these works are different. However, for the sake of illustration, a comparison with state-of-the-art approaches was conducted. It is also worth mentioning that the work reported in [43] used the same data as the case study described here but with different sampling periods (1 s in [43] and 1 min here). The washing machine and fridge are two popular multi-state devices for evaluating state-of-the-art NILM approaches, and they were used here. Table 16 compares the proposed NILM framework to some state-of-the-art NILM approaches for the washing machine. The fridge's performance in comparison with state-of-the-art approaches is presented in Table 17. In both tables, the column labeled S(s) denotes the sampling time in seconds.

**Table 16.** Washing machine performance comparison.

| Approach | S (s) | *Recall* | *Precision* | *F*1 | MAE (Wh) | SAE | EA |
|---|---|---|---|---|---|---|---|
| [27] | 1 | **1** | 0.60 | 0.70 | 118.1 | - | - |
| [41] | 1 | - | - | 0.76 | 14.42 | 0.51 | 0.74 |
| [42] | 60 | 0.86 | 0.87 | 0.86 | 8.31 | **0.01** | - |
| [43] | 1 | 0.96 | 0.96 | 0.96 | 1.64 | 0.05 | **0.93** |
| Proposed | 60 | 0.97 | **0.97** | **0.97** | **1.1** | 0.03 | **0.93** |

S: sampling frequency.

**Table 17.** Fridge performance comparison.

| Approach | S (s) | *Recall* | *Precision* | F1 | MAE (Wh) | SAE | EA |
|----------|-------|----------|-------------|-----|----------|------|------|
| [27] | 1 | 0.73 | 0.87 | 0.79 | 4.34 | - | - |
| [41] | 1 | - | - | 0.87 | 19.60 | 0.46 | 0.76 |
| [42] | 60 | 0.89 | 0.85 | 0.87 | 17.03 | 0.05 | - |
| [43] | 1 | 0.97 | **0.92** | **0.95** | 12.72 | 0.09 | 0.88 |
| Proposed | 60 | **0.98** | 0.91 | **0.95** | **4.00** | **0.01** | **0.96** |

According to the analysis of the comparison results presented in Tables 16 and 17, the proposed framework designed by MOGA performed better than the NILM method presented in [43] in terms of estimation accuracy for the fridge (96% versus 88% in [43]). The mean absolute error of the washing machine was reduced by 32% using the MOGA framework compared to the work presented in [43]. It should be noted that the proposed models designed by MOGA and the approach presented in [43] outperformed the other state-of-the-art methods referenced in Tables 16 and 17.

Since the design of a radial basis function neural network by MOGA does not require too much training data (around 8212 samples to 24,493 samples) when using a sampling interval of one minute, while achieving a similar or better performance than approaches using more training data [43] (around 1.3 million samples to 1.4 million samples), it is worth noting that resampling the data sampling rate from 1 s to 1 min did not affect the performance of the MOGA framework and significantly reduced the amount of data to be processed.

It should be noted that MOGA is a time complex process. Although it runs on a computer cluster, the training of models with data sampled at 1 min takes several hours. As the complexity of MOGA is linear with the number of samples, the use of the same time period, sampled at one second, would translate the execution time to several days. In practice, MOGA cannot cope with this large number of samples, which would have the consequence of reducing the design data, diminishing in this way the number of events that could be used in the design process.

### 4.1. Test on AMPD Public Dataset

The proposed framework was also evaluated using the public dataset AMPD [82] (Almanac of Minutely Power datasets). It records a Canadian household's water, natural gas, and electricity consumption data for two years, including electrical features such as voltage, current, active, and reactive power, collected at one-minute intervals. The data from 1 April 2012 to 30 April 2012 were considered. The models were trained using the same configuration as the proposed framework. Two appliances (fridge and clothes dryer) were considered in the experiment. The results of the test performed in the validation dataset in terms of energy estimation and device classification are presented in Table 18.

**Table 18.** Results on AMPD public dataset.

| Devices | MAE (Wh) | SAE | EA | *Recall* | *Precision* | F1 |
|---------|----------|-----|-----|----------|-------------|-----|
| Fridge | 9.13 | 0.10 | 0.90 | 0.91 | 0.93 | 0.92 |
| Clothes dryer | 3.53 | 0.01 | 0.94 | 0.99 | 1 | 0.99 |

From the results in Table 18, it can be seen that the fridge and the clothes dryer were detected with a high *F*1 score of 0.92 and 0.99, respectively. Regarding the energy estimation, the fridge was estimated with an accuracy of more than 90%, while the clothes dryer was estimated with an accuracy of 94%.

Examples of disaggregation outputs, in terms of active power for the fridge, are presented in Figure 9. Figure 10 shows the example of disaggregation outputs for the clothes dryer.

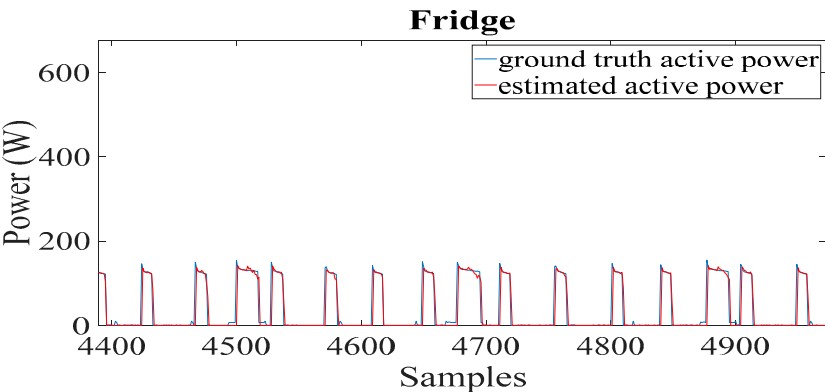

**Figure 9.** Fridge AMPD: blue, ground truth active power; red, estimated active power.

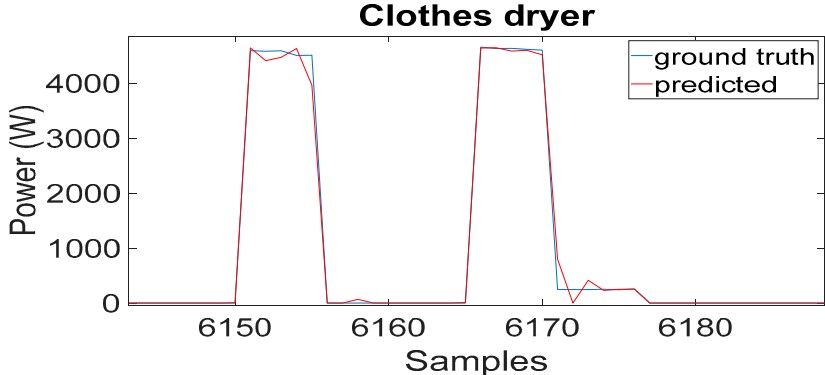

**Figure 10.** Clothes dryer AMPD: blue, ground truth active power; red, estimated active power.

There is a reasonably consistent agreement between the measured and estimated active power, as shown in Figures 9 and 10.

The results obtained with this dataset were also compared to the work proposed in [53,83]. In both works, two weeks of data were used. In [83], the authors proposed a modified cross-entropy algorithm (MCE) based on a combinatorial optimization method to classify the load states. The approach consisted of searching for the best combination of states by iteratively updating the device's operation probability to generate a load decomposition while considering the constraint of the penalty function. In [53], the authors suggested an approach based on the probability of time-segmented states. They used an affinity propagation clustering approach to extract the load power patterns, which were then used to count the probabilities of time-segmented states. After generating a range of appliance state matrices using probabilities, the function selects the most suitable matrix as the result of the appliance state detection. Table 19 compares the results of the two appliances (fridge and clothes dryer) in terms of state identification (F1) with the references [53,83].

**Table 19.** F1 comparison.

| Devices | [53] | [83] | Proposed |
|---|---|---|---|
| Fridge | 0.92 | 0.88 | 0.92 |
| Clothes dryer | 0.23 | 0.29 | 0.99 |

As illustrated in Table 19 for the fridge, the *F*1 score achieved by the proposed framework (92%) is comparable to the *F*1 score obtained by the Ref. [53], while the MCE [83] approach had the lowest score of 88%. For the clothes dryer, the proposed framework performed the best *F*1 score of 99%, while the Ref. [53] and MCE methods had the lowest

scores of 23% and 29%, respectively. We concluded from these results that the proposed framework allowed us to achieve very satisfactory results.

### 4.2. Energy Consumption Estimation in the Case Study House

The procedure used to identify the four appliances discussed in Section 4 was extended to other devices in the case study house that consume the most electricity. One month of data was considered in the model design. The results generated by the data selection algorithm and the size of the datasets for each device for classification and estimation are presented in Tables 20 and 21, respectively.

**Table 20.** Data selection results for classification models.

| Devices | CH Vertices | Training | Testing | Validation |
|---|---|---|---|---|
| AC_1 | 913 | 28,777 | 8925 | 8927 |
| AC_2 | 1170 | 28,505 | 9501 | 9503 |
| AC_3 | 569 | 26,777 | 8925 | 8927 |
| AC_4 | 799 | 26,777 | 8925 | 8925 |
| BS_1 | 714 | 26,777 | 8925 | 8927 |
| BS_2 | 638 | 26,777 | 8925 | 8927 |
| Oven | 1164 | 26,777 | 8925 | 8927 |
| DM | 993 | 24,185 | 8061 | 8063 |
| EAH_1 | 900 | 26,777 | 8925 | 8927 |
| EAH_2 | 1386 | 26,777 | 8925 | 8927 |
| EAH_3 | 1384 | 26,777 | 8925 | 8927 |

AC: Air conditioner, BS: Burner stove, DM: Drying machine, EAH: Electric air heater.

**Table 21.** Data selection results for estimation models.

| Devices | CH Vertices | Training | Testing | Validation |
|---|---|---|---|---|
| AC_1 | 1993 | 26,777 | 8925 | 8927 |
| AC_2 | 913 | 28,505 | 9501 | 9503 |
| AC_3 | 543 | 26,777 | 8925 | 8927 |
| AC_4 | 565 | 26,777 | 8925 | 8927 |
| BS_1 | 881 | 26,777 | 8925 | 8927 |
| BS_2 | 698 | 26,777 | 8925 | 8927 |
| Oven | 1249 | 26,777 | 8925 | 8927 |
| DM | 835 | 24,185 | 8061 | 8063 |
| EAH_1 | 1982 | 26,777 | 8925 | 8927 |
| EAH_2 | 1309 | 26,777 | 8925 | 8927 |
| EAH_3 | 2180 | 26,777 | 8925 | 8927 |

After one run of MOGA, the non-dominated sets were generated. The size of the non-dominated sets for the classification models and energy estimation models are shown in Table 22 for each device.

**Table 22.** Dimensions of non-dominated sets.

| Devices | Classification | Estimation |
|---|---|---|
| AC_1 | 444 | 145 |
| AC_2 | 281 | 86 |
| AC_3 | 663 | 126 |
| AC_4 | 131 | 159 |
| BS_1 | 827 | 81 |
| BS_2 | 256 | 256 |
| Oven | 539 | 213 |
| DM | 561 | 59 |
| EAH_1 | 636 | 80 |
| EAH_2 | 654 | 231 |
| EAH_3 | 768 | 443 |

Suitable models with good performance and low complexity were analyzed in the non-dominated sets. Two models (a classification and an estimation model) were selected for each device. Table 23 shows the results of the device state classification using the selected models in terms of model complexity, number of neurons in the hidden layer, number of features, recall, precision, and *F*1 score. The results in terms of device energy estimation using the selected models are shown in Table 24.

**Table 23.** Device state classification results.

| Devices | *Recall* | *Precision* | *F1* | Nb of Features | Nb of Neurons | Model Complexity |
|---------|----------|-------------|------|----------------|---------------|------------------|
| AC_1  | 0.92 | 0.94 | 0.93 | 13 | 20 | 280 |
| AC_2  | 0.97 | 0.98 | 0.97 | 5  | 30 | 180 |
| AC _3 | 0.94 | 0.90 | 0.92 | 11 | 28 | 336 |
| AC_4  | 1.00 | 1.00 | 1.00 | 9  | 11 | 110 |
| BS_1  | 0.59 | 0.81 | 0.68 | 5  | 23 | 138 |
| BS_2  | 0.86 | 0.97 | 0.91 | 12 | 29 | 377 |
| Oven  | 0.80 | 0.97 | 0.88 | 19 | 29 | 580 |
| DM    | 0.96 | 0.98 | 0.97 | 17 | 29 | 522 |
| EAH_1 | 0.98 | 0.95 | 0.96 | 4  | 30 | 150 |
| EAH_2 | 0.91 | 0.89 | 0.90 | 9  | 30 | 300 |
| EAH_3 | 0.74 | 0.76 | 0.75 | 9  | 18 | 180 |

**Table 24.** Energy estimation results.

| Devices | MAE (Wh) | SAE | EA | Nb of Features | Nb of Neurons | Model Complexity |
|---------|----------|-----|----|----------------|---------------|------------------|
| AC_1  | 191.0 | 0.011 | 0.87 | 14 | 18 | 270 |
| AC_2  | 11.0  | 0.002 | 0.97 | 3  | 17 | 68  |
| AC _3 | 11.0  | $4 \times 10^{-4}$ | 0.98 | 7  | 10 | 80  |
| AC_4  | 9.0   | 0.003 | 0.97 | 15 | 18 | 288 |
| BS_1  | 90.0  | 0.004 | 0.86 | 3  | 20 | 80  |
| BS_2  | 37.0  | 0.002 | 0.90 | 3  | 4  | 16  |
| Oven  | 43.0  | 0.005 | 0.73 | 22 | 19 | 437 |
| DM    | 0.009 | $4 \times 10^{-4}$ | 0.98 | 3  | 9  | 36  |
| EAH_1 | 28.0  | 0.010 | 0.95 | 12 | 13 | 169 |
| EAH_2 | 10.0  | 0.004 | 0.99 | 3  | 12 | 48  |
| EAH_3 | 14.0  | 0.006 | 0.87 | 16 | 6  | 102 |

Analyzing the classification results presented in Table 23, we observed an excellent *F*1 score of 100% for the air conditioner (AC-4), whereas the lowest *F*1 scores were observed in the classification of the burner stove (BS-1: 68%) and the electric air heater (EAH-3: 75%). In the first case, a higher sampling frequency should be used, while in the second case, the device is rarely activated and there is a lack of identifiable data. The other appliances were classified with good *F*1 scores ranging from 88% to 98%. The results reported in Table 24 show that the electric air heater (EAH-2) was estimated with an excellent estimation accuracy (EA) of 99%. The lowest performance was observed in the disaggregation of the oven (73%). This is due to the fact that the oven has several different operating modes, with different consumptions together with a large range of temperatures. The other devices were estimated with good estimation accuracy ranging from 86% to 98%.

Once the models were designed, the energy consumption of certain appliances in the case study house was estimated using these models. The aggregated data from January 2022 were considered in the experiment. Table 25 shows the results of the disaggregation in terms of appliance energy consumption. Since the aggregated data were measured by the EM340 three-phase smart meter, the distribution of appliance consumption by phase is presented.

**Table 25.** Distribution of energy consumption in the case study house.

| Phase | Appliances | E (kWh) |
|---|---|---|
| I | Air conditioner (AC1) | 11.97 |
| | Air conditioner (AC2) | 91.98 |
| | Burner stove (BS1) | 14.42 |
| | Electric air heater (EAH1) | 24.56 |
| | Electric air heater (EAH3) | 28.40 |
| | Electric air heater (EAH2) | 95.52 |
| II | Drying machine (DM) | 13.36 |
| | Electric water heater (EWH) | 114.54 |
| | Fridge | 17.80 |
| | Washing machine (WM) | 13.73 |
| | Swimming pool pump (SPP) | 153.14 |
| | Oven | 19.22 |
| III | Air conditioner (AC3) | 16.32 |
| | Air conditioner (AC4) | 71.74 |
| | Burner stove (BS2) | 19.27 |

Analyzing the disaggregation results reported in Table 25, it can be seen that the highest energy consumption in the month of January 2022 was assigned to the swimming pool pump (153.14 kWh) followed by the electric water heater (114.54 kWh). The electric air heater (EHA2) and air conditioners (AC2 and AC4) consumed approximately 11.97 kWh, 91.98 kWh, and 71.74 kWh respectively. The consumption of the other appliances was estimated between 4.22 kWh and 28.40 kWh. Figure 11 presents pie charts summarizing the distribution of electricity consumption in the case study house in the month of January 2022.

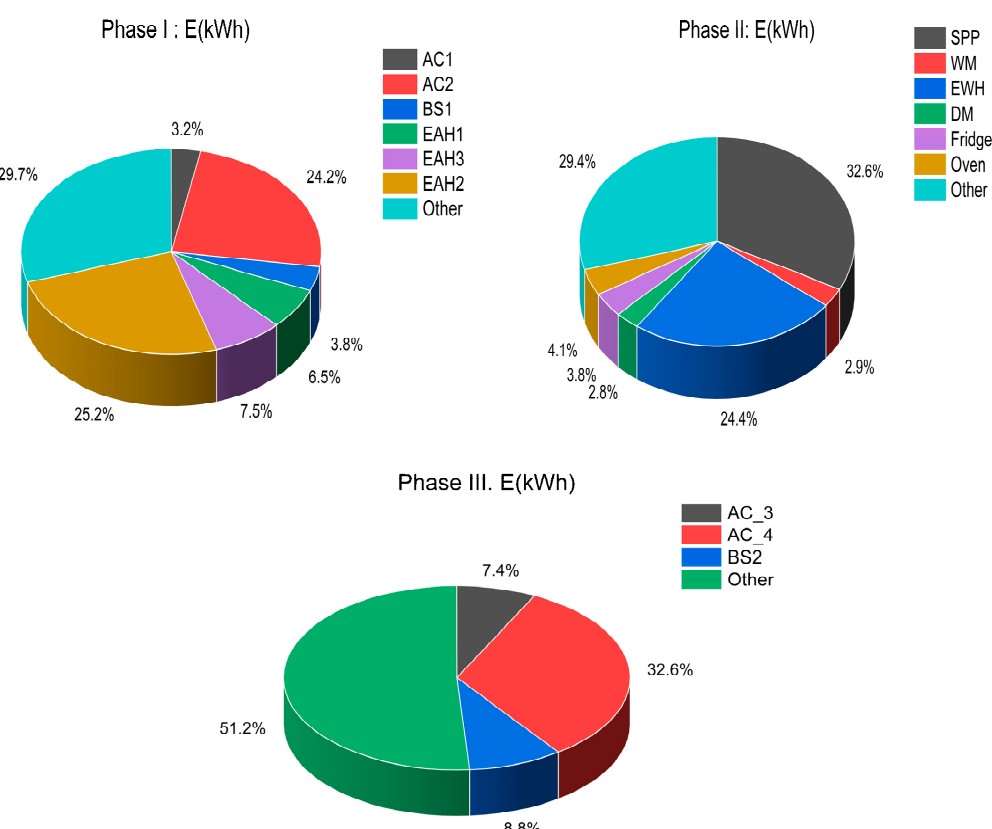

**Figure 11.** Distribution of electricity consumption.

The total consumption of the house, during the month of January 2022, was 1070 kWh, divided into 380, 470, and 220 kWh for Phases 1 to 3. The appliances considered in this section account for 66% of the total monthly consumption. Besides being responsible for most of the electric energy consumption in the household, most of these appliances are schedulable, meaning that their operation can be changed without causing much trouble for the occupants. This is the case with HVAC appliances (AC 1-4, EAH 1-3, WM, DM, SPP, and, to a less extent, EWH). These appliances account for 60% of the house's total consumption, which means that there is considerable flexibility in shaping the house electric profile to meet energy management goals. As the house in question has PV panels and electric storage, the online operation of some of these electric appliances is taken into account in our proposed model-based predictive control of HEMS [84].

## 5. Conclusions

In this study, a low complexity NILM framework based on radial basis function neural networks designed by a multi-objective genetic algorithm (MOGA) was proposed for energy disaggregation. Despite reducing the data sampling from one second to one minute to allow for the use of low-cost meters, the reduction of design time, and the employment of low complexity models, the proposed technique presented an excellent ability to disaggregate the usage of devices.

A comparative analysis of other computational intelligence classifiers for non-intrusive load monitoring, using the same data, showed that the proposed framework obtained the best experimental results in terms of appliance identification. The comparison with other state-of-the-art methods, both using different data and common data, highlighted the efficiency of the proposed framework in achieving the best estimation of the energy consumed by each device in the house.

The proposed NILM technique was used to disaggregate one month of consumption of the house, and it was able to identify the operation of appliances accounting for 2/3 of the electric consumption. It allowed us also to recognize that around 60% of the consumption was related to schedulable appliances, therefore allowing an additional flexibility for the HEMS available in the residence.

Future work will be devoted to the question of the transferability of the proposed framework to other houses, as well as its integration in the house HEMS. One additional advantage of using very simple models is that their real time execution is in the order of a few milliseconds in standard computer architectures, which allows an edge implementation.

**Author Contributions:** Conceptualization, I.L. and A.R.; methodology, I.L., M.d.G.R. and A.R.; software, I.L. and A.R.; validation, I.L., M.d.G.R. and A.R.; formal analysis, I.L., I.G., M.d.G.R., H.E.F., S.D.B. and A.R.; investigation, I.L., M.d.G.R. and A.R.; resources, M.d.G.R. and A.R.; data curation, I.L. and A.R.; writing—original draft preparation, I.L., M.d.G.R. and A.R.; writing—review and editing, I.L., I.G., M.d.G.R., H.E.F., S.D.B. and A.R.; supervision, A.R., H.E.F., and S.D.B.; project administration, M.d.G.R. and A.R.; funding acquisition, M.d.G.R. and A.R. All authors have read and agreed to the published version of the manuscript.

**Funding:** This research was funded by Programa Operacional Portugal 2020 and Operational Program CRESC Algarve 2020, grant numbers 39578/2018 and 72581/2020. Antonio Ruano also acknowledges the support of Fundação para a Ciência e Tecnologia, grant UID/EMS/50022/2020, through IDMEC under LAETA.

**Data Availability Statement:** Data supporting reported results can be found at https://csi.ualg.pt/nilmforihem, (accessed on 1 November 2022).

**Conflicts of Interest:** The authors declare no conflict of interest.

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
