# Peer review of "Energy Disaggregation Using Multi-Objective Genetic Algorithm Designed Neural Networks"

_energies, doi:10.3390/en15239073_

Round 1

Reviewer 1 Report

After a thorough review of the paper titled " Energy Disaggregation using Multi-Objective Genetic Algorithm designed Neural Networks", it has been found that, no doubt, that the proposed work is interesting in terms of estimating residential energy usages. However, the methodology, findings and conclusions are not well presented as per the Quantification perspective of energy use estimates in residential buildings. Several main points that were supposed to raise in this study, are as follows.

- Abstract: Please ensure that the abstract has the following elements: 1-2 sentences on the context and the need for the study; 1-2 sentences on the methodology; the majority of the abstract on the actual results of the study; 1-2 sentences on key conclusions and recommendations. 

- Manuscript structure: Please structure the manuscript and title the sections as follows: 1. Introduction; 2. Methodology [2.1. Problem statement, 2.2. Data selection, 2.3. Dataset, 2.4. Data preprocessing, 2.5. proposed classification method (2.5.1. proposed RBFNN-MOGA method; and 2.5.2. Other implemented classification methods), 2.6. Performance metrics]; 3. Results; 4. Discussion; 5. Conclusions. All other sections must be integrated within one of these sections. If you prefer, you can merge the results and discussion sections into one. The conclusions section should not have any subtitles or references. All sections and subsections must be numbered.

 - The study novelty: The objective of this study is still unclear, for example, the authors generally mentioned in the introduction, as follows “In this paper, a NILM framework based on a low frequency sampling rate, allowing therefore the use of low-cost meters, and employing low complexity shallow neural network models is proposed.  The following are the key contributions of this paper:”, however, questions regarding the importance of this work, the problem that has been addressed by this work, all of these significant aspects remained absent. The introduction did not strictly address the problem, i.e., what is the importance of the proposed methodology in light of previous studies? In other words, what is the research gap that a proposed method addresses if compared to the methods used in previous studies? Unfortunately, this question is still absent.

- Methodology: This section is so long? Also, the methodology described in this section is not adequate organized, this makes the reader is confused. Please organize this section to present the proposed methodology in a smooth way.  Moreover, the figures 1-3, are unclear, I don’t know why? The figure must be clear to be able to communicate the goal that was set.  Please redraw these figures in a smooth way. Besides, in subsection of performance metrics (i.e., Accuracy metrics), what the units of measurement used in this subsection? Please ensure that this metrics should be clear.

- Results: Tables 2-25, are not clear, I do not know why? The quality of this paper is not in the number of tables presented, the quality is in the clear and understandable content of these tables. Please ensure that these tables include the suitable title, unit, and right results to show the ability of proposed method in the research problem processor/wizard.   Moreover, subsection of 3.1. Data preprocessing, should be moved to the section of methodology.

- Conclusions: Please ensure that this section only present the key findings. Limitations of the study and potential future development of this research should also be included in the manuscript.

Please include a point-by-point reply to the above comments, alongside the reply to the reviewers' comments. Please detail the revisions that have been made, citing the line numbers and the changes made. 

Author Response

Thank you very much for your comments which have clearly improved the paper

Reviewer 2 Report

The paper is of interest, is well written and structured and contains a lot of useful information. I have only a few comments to make:

1. Moving from 1 sec to 1-min resolution for a more light-weight approach is clear. What I would like to see is some kind of analysis about the computational burden with 1-sec and 1-min data to understand how big is the impact?

2. A few appliances are examined and the accuracy metrics are good. What would be the response of the proposed implementation in case of more appliances being added? water boilers, other types of dryers, maybe less energy intensive appliances?

3. Where this solution is expected to "run"? on the cloud or maybe even on the edge since this is a light-weight version? What are the HW needs?

4. Although the references list is already rich with plenty of papers and some recent ones also, I would recommend that the authors include the following 2 papers also. Both refer to scalable light-weight NILM solutions based on different resolutions and for real-time this time that would further enrich the literature part of the paper.  A. Athanasiadis, Christos L., Theofilos A. Papadopoulos, and Dimitrios I. Doukas. "Real-time non-intrusive load monitoring: A light-weight and scalable approach." Energy and Buildings 253 (2021): 111523. and B. Athanasiadis, Christos, et al. "A scalable real-time non-intrusive load monitoring system for the estimation of household appliance power consumption." Energies 14.3 (2021): 767.

Author Response

(The authors gave the same response as above.)

Round 2

Reviewer 1 Report

Compared to the first draft of this manuscript, this revised manuscript has been improved by addressing the reviewers' comments.

Reviewer 2 Report

no further comments